# Domestication Gene *Mlx* and Its Partner *Mondo* Are Involved in Controlling the Larval Body Size and Cocoon Shell Weight of *Bombyx mori*

**DOI:** 10.3390/ijms25063427

**Published:** 2024-03-18

**Authors:** Xiaoxuan Qin, Liang Jiang, Ping Zhao, Ying Lin, Yi Zhang, Qingyou Xia

**Affiliations:** 1Integrative Science Center of Germplasm Creation in Western China (Chongqing) Science City, Biological Science Research Center, Southwest University, Chongqing 400715, China; qinxiaox@foxmail.com (X.Q.); jiangliang@swu.edu.cn (L.J.); zhaop@swu.edu.cn (P.Z.); ly908@swu.edu.cn (Y.L.); 2Key Laboratory for Germplasm Creation in Upper Reaches of the Yangtze River, Ministry of Agriculture and Rural Affairs, Chongqing 400715, China; 3Chongqing Key Laboratory of Sericultural Science, Chongqing Engineering and Technology Research Center for Novel Silk Materials, Southwest University, Chongqing 400715, China; 4State Key Laboratory of Resource Insects, Institute of Sericulture and Systems Biology, Southwest University, Chongqing 400715, China

**Keywords:** silkworm, domestication, transcription factor, *Mlx*, *Mondo*

## Abstract

*Bombyx mori* was domesticated from *Bombyx mandarina*. The long-term domestication of the silkworm has brought about many remarkable changes to its body size and cocoon shell weight. However, the molecular mechanism underlying the improvement in the economic characteristics of this species during domestication remains unclear. In this study, we found that a transposable element (TE)—Bm1—was present in the upstream regulatory region of the *Mlx* (Max-like protein X) gene in wild silkworms but not in all domesticated silkworms. The absence of Bm1 caused an increase in the promoter activity and mRNA content of *Mlx*. Mlx and its partner Mondo belong to the bHLHZ transcription factors family and regulate nutrient metabolism. RNAi of *Mlx* and *Mondo* decreased the expression and promoter activity of glucose metabolism-related genes (*trehalose transport* (*Tret*), *phosphofructokinase* (*PFK*), and *pyruvate kinase* (*PK*)), lipogenic genes (*Acetyl-CoA carboxylase* (*ACC*) and *fatty acid synthase* (*FAS*)), and glutamine synthesis gene (*Glutamine synthase 2,* (*GS2*)). Furthermore, the transgenic overexpression of *Mlx* and *Mondo* in the fat body of silkworms increased the larval body size, cocoon shell weight, and egg number, but the silencing of the two genes resulted in the opposite phenotypes. Our results reveal the molecular mechanism of *Mlx* selection during domestication and its successful use in the molecular breeding of *Bombyx mori*.

## 1. Introduction

The domesticated silkworm, *Bombyx mori*, is the only truly domesticated lepidopteran insect [1,2], and has important economic value. The gene recombination rate and polymorphism of domesticated silkworms have decreased, and the homozygosity of domesticated silkworms has increased during more than 5000 years of purposeful selection [3,4]. Long-term domestication leads to remarkable changes in body color and size [5], however, the molecular mechanism responsible for these improvements remains unknown. Many works have revealed that transposable elements (TEs) play important roles in building eukaryotic genome evolution and function, after McClintock B. proposed for the first time that transposable genes control the color of maize grains [6,7,8]. TEs constitute an important part of the genome of most multicellular eukaryotes and can profoundly affect their evolution [9]. For example, in plants, the insertion of hopscotch transposon causes corn straw to change from multiple branches to single branches [10]. Blood orange takes advantage of a Copia retrotransposon adjacent to a gene-encoding Sicilian Ruby-related anthocyanin [11]. Many studies on TE movement and function in metazoans have been documented in *Drosophila* [12,13]. In animals, TE insertions are associated with body size and alteration in the hair color of dogs and cats [14,15,16,17]. In *Bombyx mori*, the Taguchi transposon retained during domestication is located in the regulatory region of the ecdysone oxidase gene (*EO*) and enhances its expression, inhibiting the premature maturation of larva [18]. However, more cases of TE in *Bombyx mori* domestication are still needed.

Sugar provides energy for the normal physiological activities of multicellular organisms and is the basis for the survival and reproduction of organisms. There are other regulatory factors along with the insulin pathway involved in the effect of sugar on organisms and cells [19]. Some studies show that Mondo-Mlx can sense phosphorylated hexoses such as glucose-6-phosphate in cells [20,21,22]. The transcription factors Mlx and Mondo belong to the bHLHZ family [23,24,25,26]. MondoA and ChREBP (carbohydrate response element-binding protein, MLXIPL, or MondoB) are paralogs in mammals in which MondoA is mainly expressed in the skeletal muscle [24], while ChREBP plays a role in the liver, adipose tissue, and pancreatic β cells [26,27,28]. In invertebrates, there is generally a single ortholog called Mondo.

Mondo-Mlx binds to the ChoRE (carbohydrate response element) motif as heterodimers to regulate most of the global glucose-induced transcriptional responses and many of their target genes involved in carbohydrate, lipid, and amino acid metabolism [29,30,31,32,33,34,35,36]. These target genes include *Phosphofructokinase 2* (*PFK2*), *Aldehyde dehydrogenase type III* (*Adh III*), *Glucose-6-phosphate dehydrogenase* (*G6PD*), *FAS*, *ACC*, *GS2,* and so on in flies [29,30,31,32,33,34,36,37]. Meanwhile, Mondo-Mlx regulates sugar transport GLUT and the carbohydrate digestion gene *Amy-p* [33]. In addition, Mondo-Mlx can also indirectly regulate transcription factors to modulate metabolic homeostasis. The Krüppel-like transcription factor Cabut, a Mondo-Mlx target, inhibits *pepck* and the circadian cycling of metabolic genes to coordinate energy metabolism and the circadian clock in response to sugar sensing [31,38]. Moreover, Mondo-Mlx regulates the Activin ligand Dawdle and Gli-Similar transcription factor Sugarbabe to mediate organismal sugar sensing [33].

In order to clarify the biological roles of Mondo-Mlx, many studies have been carried out using loss-of-function organizations or individuals. The lifespan of C. elegan is shortened by a lack of mml-1(Mondo) or mxl-2(Mlx) [39]. Mlx null mutant flies display dramatic sugar intolerance, including delayed pupation and reduced pupal survival on a high-sugar diet [31]. ChREBP−/− mice have difficulty surviving on a high-sugar diet [40]. ChREBP/MondoA-Mlx is associated with metabolic disease disorders in humans. A novel short ChREBP isoform (ChREBPβ) is related to circulating triglyceride levels, and decontrolled ChREBP is involved in severe obesity in adipose and liver tissue [41,42,43]. In other reports, ChREBP plays a key role in maintaining fructose tolerance in liver and intestine tissues [44,45]. Moreover, ChREBP/MondoA-Mlx is involved in regulating metabolic processes related to the proliferation of cancer cells and tumorigenesis by deregulating the Myc oncogene [46,47,48,49].

The absorption and utilization of carbohydrates by organisms require the coordination of a variety of biological processes, including gene regulation and protein–protein interactions in metabolic networks. The effective utilization and transformation of nutrients in mulberry leaves by silkworms not only form the material basis for growth and development but also influence the yield of cocoons and eggs in silkworms. Silkworms absorb carbohydrates from mulberry leaves to consume them as energy and synthesize storage carbohydrates, such as glycogen, lipids, and chitin. The absorption and utilization of nutrients in mulberry leaves are important traits in sericulture production, and their regulation network exists under the pressure of artificial selection. Previous studies have shown that the key transcription factor Mlx regulating carbohydrate metabolism, is located in the genomic regions of domesticated selection signals, indicating that *Mlx* may have been selected [5,50,51].

In this study, we explored the molecular adaption of the transcription factor Mlx in the process of silkworm domestication, and investigated the physiological functions of the Mondo-Mlx complex in the fat body of silkworms. We observed the presence of a TE (Bm1) in the upstream regulatory region of *Mlx* in wild silkworms, contrasting its total absence in the domesticated silkworm strains. The absence of Bm1 in the upstream regulatory region may be involved in inhibiting *Mlx*, probably due to removing the repression of Cabut. The upstream region of *Mlx* is selected during domestication and TE is removed in domesticated silkworms. Mlx and Mondo also regulate metabolic genes and transcription factors in silkworm. We generated transgenic knocking down and overexpression strains in silkworm to better study the physiological roles of Mondo-Mlx. Mondo-Mlx is related to the individual size of silkworms, the quality of cocoons, and the number of silkworm eggs. Overexpressing *Mlx* or *Mondo* can increase the content of triglyceride, increase the mass of the fat body, and thus improve the weight of silkworms and economic characteristics. Interfering with *Mlx* or *Mondo* reduces the weight of silkworms and their quality of economic characteristics. Our works suggest that the domestication gene *Mlx* and its partner *Mondo* can be used as molecular targets to promote growth and production in silkworm breeding.

## 2. Results

### 2.1. Domestication Selection Analysis of a TE Deletion in the Upstream Regulatory Region of Mlx Gene in Domesticated Silkworm

According to the analysis of the resequencing results of domesticated and Chinese wild silkworm strains, it was predicted that *Mlx* is one of the domestication candidate genes [5,50,51]. To study the differences in and the role of the *Mlx* gene between domesticated and wild silkworms, further research was carried out. We amplified the upstream region of the *Mlx* translation start codon (~1000 bp) from four domesticated and two wild strains. There was a single fragment (~1000 bp) in all domesticated silkworms, however, another longer fragment (~1500 bp) appeared in wild silkworms (Figure 1a). We found that there was no short band in lane 2 of WS1 and no long band in lane 1 of WS2. We speculate that if the genome of wild silkworm is homozygous, there is only a long band, and when the genome is heterozygous, short and long bands will appear. We sequenced the short and long fragments from these six strains. The major difference between the sequences from domesticated and wild silkworms is the existence of an approximately 450 bp (in W1 and W2) fragment. This 450 bp fragment was predicted to be a short interspersed nuclear element (SINE), Bm1 (Appendix A), located in ~380 bp upstream of the transcription start site (TSS) of the *Mlx* gene (Figure 1b). The amplified sequences are shown in Appendix A. The results shown that a retrotransposon was absent in the upstream regulating region of the *Mlx* gene in domesticated silkworms. Thus, the retrotransposon deletion might be adaptive during domestication.

We searched for the domestication selection signatures in the upstream region of *Mlx* translation start codon to determine the deletion of TE in domesticated silkworms (eighteen domesticated and four wild silkworm strains, Appendix A). In this region, the polymorphism was lower in the domesticated strains (0.00842 vs. 0.01751), and the polymorphism was also lower in the non-TE sequences (0.02301 vs. 0.06996) (Table 1). This shows that the genetic diversity of domesticated silkworms was simpler than that of wild silkworms in this region. Meanwhile, Tajima’s D value was negative (−2.62506) in domesticated silkworms, but it was positive in wild silkworms (0.34000) (Table 1). This indicates that this region was under pressure from directional selection and selective sweep. This evidence indicates that the region that changes from wild silkworms to domesticated silkworms is under pressure from directional selection, and the deletion of Bm1 is the result of artificial domestication selection.

### 2.2. Retrotransposon Bm1 Is Involved in Modulating the Expression of Mlx

To confirm the fact that Bm1 deletion affected the expression of nearby genes, we compared the expression of *Mlx* in the fat body of domesticated and wild silkworms. We found that the expression of *Mlx* was much higher in domesticated silkworms than wild silkworms (Figure 1c). Moreover, the expression of some metabolism-related genes and factors, such as *ACC*, *FAS* and *Cabut*, was higher in domesticated silkworms, which could be regulated by Mlx (Appendix A). Then, we constructed luciferase reporter systems including *Mlx* gene upstream sequences without Bm1 (-Bm1, in domesticated silkworm) or with Bm1 (+Bm1, in wild silkworm). The luciferase activity is higher in the sequences without Bm1 than in those with Bm1 (Figure 1d). To verify the universally and broad transcriptional repression of Bm1, we utilized a luciferase reporter system containing Bm1 before the *Hsp70* promoter. The absence of Bm1 was associated with higher transcriptional activity of the *Hsp70* promoter, while its presence was associated with lower *Hsp70* promoter activity (Figure 1e). Moreover, this association was also supported by elevated transcription and translation of DsRed (Figure 1f–h). The results shows that the presence of Bm1 was associated with transcriptional repression.

We predicted the transcription factors binding to the Bm1 sequence to further clarify how Bm1 is associated with transcriptional repression. We found six GC-rich motifs between the 25th and 35th bp of Bm1, representing a potential KLF10 (or SP1) transcription factor binding site (Appendix A). The KLF10/SP1 transcription factor is a kind of transcription factor containing zinc finger structures in mammals [52,53,54]. KLF10/SP1 can recruit the HDAC1-Sin3A complex and bind to GC-rich DNA region, leading the chromatin concentration to inhibit the transcription of downstream genes [55,56,57,58,59]. The homologous protein of the KLF10/SP1 transcription factor is Cabut in flies [38,60,61]. The luciferase activity was increased after interfering with *Cabut* via RNAi when Bm1 was present, meaning that the transcriptional repression associated with Bm1 was weakened (Figure 1i). In contrast, the overexpression of *Cabut* produced the opposite result (Figure 1j). Acetylation and deacetylation modifications of histones can affect gene expression. HAT (histone acetyltransferase) acetylates the histones, forming an “open” chromatin structure to facilitate transcription. On the contrary, HDAC (histone deacetylase) deacetylates histones and forms a closed structure in chromatin, leading to gene silencing [62,63,64,65,66,67,68,69,70]. Trichostatin A (TSA) is a commonly used broad-spectrum inhibitor of HDAC family. TSA can strongly and selectively inhibit HDAC. The inhibition of HDAC by TSA can increase histone acetylation levels and lead to increased transcription levels of some genes [71,72]. After treatment with TSA, the lesser repression in the presence of Bm1 in the upstream regulatory region supports our hypothesis that additional deacetylation in Bm1 affects the gene expression level (Figure 1k).

The above results show that *Mlx* and some metabolism-related genes were more highly expressed in domesticated than in wild silkworms. The presence of Bm1 in the regulatory region of *Mlx* is associated with repressing *Mlx* expression. This repression may be mediated by Cabut binding and histone deacetylation.

### 2.3. Mondo-Mlx Regulates the Expression of Genes Related to Trehalose Transport and Glycolysis

Trehalose is the blood glucose of the silkworm, and its direct energy source [73]. The trehalose transporter Tret can transfer trehalose to maintain the homeostasis of blood glucose. After interfering with *Mlx* or *Mondo* (Figure 2a,b), we found that the expression of *Tret* decreased (Figure 2c), and the transcriptional activity of the *Tret* promoter decreased (Figure 2d). ChIP-PCR results showed that Mlx or Mondo could bind to the *Tret* promoter to directly activate it (Figure 2e,f). This indicates that Mondo-Mlx is involved in the metabolic homeostasis of blood glucose by regulating *Tret* and improving sugar tolerance.

Previous research with mammals and flies has demonstrated that the downstream target genes of ChREBP/MondoA-Mlx involve many intermediate enzymes of the glycolysis process [29,32,33,34]. In silkworm, the glycolytic pathway is like that of flies and mammals. After knocking down *Mlx* and *Mondo*, the expression of *PFK* (Figure 2g) and *PK* (Figure 2k) decreased, which were the key rate limiting enzymes in the glycolysis process. The transcriptional activities of the *PFK* (Figure 2h) and *PK* (Figure 2l) promoter were inhibited with a decrease in the expression of *Mlx* and *Mondo*. ChIP-PCR was used to verify that Mlx and Mondo directly regulate *PFK*. The results shows that Mlx and Mondo can directly bind to the promoter of *PFK* (Figure 2i,j), indicating that Mondo-Mlx is involved in regulating glycolysis by activating *PFK* and *PK*.

### 2.4. Direct Regulation of Lipogenic Genes by Mondo-Mlx

As one of the three major nutrient types, fatty acids can not only provide energy for organisms but are also the raw materials for synthesizing other substances. ChREBP-Mlx in mammals and flies regulate transcription of the lipogenic genes *ACC* and *FAS* [29,30,31,33,36,37]. When RNAi was performed for *Mlx* or *Mondo* in BmNs cells, the expression of *ACC* was decreased, which is the rate-limiting enzyme in the fatty acid synthesis pathway (Figure 3a). The result of dual-luciferase activity detection showed that the transcriptional activity of the *ACC* promoter was repressed when the expression of *Mlx* and *Mondo* was knocked down (Figure 3b). Next, we checked that Mlx and Mondo regulated *ACC* directly via ChIP-PCR. The results reveal that Mlx and Mondo can specifically immunoprecipitated the ChoRE motif of the *ACC* promoter, demonstrating that Mlx and Mondo can directly bind to the *ACC* promoter (Figure 3c,d). Furthermore, EMSA was carried out to prove the direct binding for Mlx in vitro. The result indicates that Mlx can bind to the biotin probe in a dose-dependent manner (Figure 3e). This binding can be competitively inhibited by unlabeled cold probes, and a mutated cold probe cannot compete with this binding (Figure 3e). This indicates that Mondo-Mlx regulates *ACC* gene by specifically binding to the *ACC* promoter.

Meanwhile, the expression of *FAS*, another rate-limiting enzyme gene in fatty acid synthesis, was reduced after the RNAi of *Mlx* or *Mondo* (Figure 3f). Similarly, the dual- luciferase reporter assay confirmed that the transcription activity of the *FAS* promoter was decreased (Figure 3g). ChIP-PCR revealed that Mlx or Mondo could bind to the *FAS* promoter (Figure 3h,i). These data together demonstrate that Mondo-Mlx activates the expression of the lipogenic genes *ACC* and *FAS* in *Bombyx mori*.

### 2.5. Mondo-Mlx Regulates Directly the Expression of Glutamine Synthase 2

The non-essential amino acids glutamate and glutamine play an essential role in amino acid and energy metabolism [74]. Mattila J. found that glutamate/glutamine biosynthetic genes were induced by Mlx in high sugar feeding, and the promoter of the *GS2* was bound by Mlx in flies [33]. When interfering with Mlx or Mondo in BmNs cells, the expression of GS2 was decreased (Figure 4a). The dual-luciferase reporter assay showed that *GS2* promoter activity is regulated by Mlx and Mondo (Figure 4b). After the overexpression of *Mlx* or *Mondo-Myc*, the ChIP-PCR results showed that genome fragments could be immunoprecipitated by Mlx or Myc antibodies, respectively, and amplified a single specific band (Figure 4c,d), indicating that Mondo-Mlx can specifically bind to the *GS2* promoter. The above results show that Mondo-Mlx can directly regulate the transcription of *GS2*, thereby regulating the synthesis of glutamine. Taken together, the biosynthesis of glutamine needs to be coordinated with sugar metabolism in order to maintain the optimal growth state of silkworm. Mondo-Mlx is a central regulator in this process.

### 2.6. Mondo-Mlx Controls Cabut, Cycle and FAMeT5 Expression

In *Drosophila*, Mondo-Mlx activates the expression of Krüppel-like transcription factor Cabut [31,38], which can inhibit the expression of lipolysis, glycogenesis and glycerol metabolism-related genes such as *Brummer* (*Bmm*), *Phosphoenolpyruvate carboxykinase* (*PEPCK*), *Fructose-1,6-bisphophase* (*FBP*), and *Glycerol kinase* (*Gyk*) [38]. In addition, *Cabut* is regulated by the core circadian clock protein Cycle [38] which can link the metabolism regulation with the circadian rhythm. In BmNs cells, Mlx and Mondo were knocked down. The qRT-PCR result showed that the expression of *Cabut* decreased with knocking down *Mlx* or *Mondo* (Figure 5a), and the transcriptional activity of *Cabut* promoter was reduced when Mlx or Mondo was decreased (Figure 5b). Moreover, ChIP-PCR analysis revealed that Mondo-Mlx can bind the *Cabut* promoter to directly regulate Cabut (Figure 5c,d) in silkworms.

In order to find new downstream target genes in silkworms, *Mlx* or *Mondo* was knocked down in BmNs cells, and samples were sent for transcriptome sequencing. The results show that the knock down of *Mlx* has 89 upregulated genes and 217 downregulated genes (Appendix A), while the RNAi of *Mondo* has 85 upregulated genes and 228 downregulated genes (Appendix A). There are 49 genes co-activated and 116 genes co-inhibited by Mlx and Mondo (Appendix A, Appendix A). We found that the core circadian clock gene *Cycle* (Appendix A, BGIBMGA003869) is one of the genes regulated by Mondo. When knocking down Mondo via shRNA, the expression and promoter transcriptional activity of *Cycle* was reduced (Figure 5e,f). Farnesoic acid O-methyltransferase (FAMeT) is considered to play a catalytic role in the biochemical process of farnesoic acid (FA) being methylated to methyl farnesoate (MF) in arthropods [75,76,77]. In our results, we found that *FAMeT5* (Appendix A, BGIBMGA006318) was also a target gene of Mlx and Mondo, and the transcription of *FAMeT5* was regulated by Mlx and Mondo (Figure 5g–i). In summary, we demonstrated that Mondo-Mlx regulated two transcription factor genes (*Cabut* and *Cycle*) and a potential enzyme gene *FAMeT5*.

### 2.7. Manipulating the Expression of Mondo-Mlx Alters Body Size and Cocoon Weight

In order to investigate the biological function of Mondo-Mlx in individuals, we employed transgenic interfered and overexpressed vectors with a piggyBac (pBac) transposon [78], and we obtained overexpressed and knocked down transgenic silkworms (Appendix A). With the overexpression of *Mlx* or *Mondo* in the fat body, their target genes were upregulated (Appendix A). According to the observations and phenotype statistics, the larva body, pupal and cocoon size of the overexpression strain became larger than those of WT (Figure 6a–d and Figure 7a–d). The cocoon shell weight was increased in the overexpression strain (Figure 6e and Figure 7e). However, the cocoon shell ratio displayed no significant differences (Figure 6f and Figure 7f). After spawning, the number of eggs in the overexpression strain was raised (Figure 6g,h and Figure 7g,h). In addition, the cell size of the fat body was larger in the overexpression strain (Figure 6i,j and Figure 7i,j), and we found that the triglyceride content increased significantly the overexpression strain (Figure 6k and Figure 7k). In sum, Mondo-Mlx can promote fat accumulation, increase fat mass, and improve the economic characteristics of silkworms.

Meanwhile, when *Mlx* or *Mondo* was knocked down in the fat body, target genes of Mondo-Mlx were repressed (Appendix A), and the larvae became smaller in the knocked down strain (Figure 8a,b and Figure 9a,b). The pupae and cocoons also became significantly smaller (Figure 8c and Figure 9c). The weight of the pupa and cocoon shell decreased (Figure 8d,e and Figure 9d,e), but the cocoon shell ratio displayed no significant difference (Figure 8f and Figure 9f). The number of eggs was reduced in knocked down strain (Figure 8g,h and Figure 9g,h). Moreover, the fat body cell size of the knocked down strain was smaller than that of WT (Figure 8i,j and Figure 9i,j), and the triglyceride content of fat body cells decreased in the knocked down strain (Figure 8k and Figure 9k). In sum, the knock down of Mondo-Mlx can decrease fat mass accumulation and negatively impact the economic characteristics of silkworms.

## 3. Discussion

In the long-term process of domestication and selection, major changes have taken place in the biological processes of domesticated silkworms, including energy metabolism. Energy metabolism is the basic physiological process of silkworms and is an important characteristic that affects the production of sericulture. Mlx, a factor of the regulatory network for energy metabolism, is one of 354 candidate genes located in genomic regions of selective signals [50], and is under pressure from artificial domestication and selection in *Bombyx mori*. In this study, we found the domesticated molecular mechanism of Mlx. An adaptive TE (Bm1) deletion in the upstream regulating region of the *Mlx* gene can increase the expression of *Mlx* and the subsequent metabolism regulating genes in domesticated silkworms. The transcriptional repression associated with Bm1 could be partly mediated by Cabut and histone deacetylation. DNA methylation of Bm1 may also inhibit *Mlx* in wild silkworms because a CpG island exists in Bm1 (Appendix A). Based on the above results, we hypothesize that Mlx is involved in improving the individual growth and economic characteristics of silkworms during domestication. Moreover, Bm1 has multiple copies in the whole genome, and some are also inserted adjacent to the regulatory region of coding proteins [79,80], which may also alter the expression of nearby genes. Although this study can provide an example of Bm1 affecting nearby genes, the exact effects of Bm1 relative to other nearby genes must be examined further.

Previous studies have determined that Mondo-Mlx has preserved and physiological significance in model animal mutants (*Drosophila* and *Rattus*) [31,40]. Much evidence has also proven that the physiological function of Mondo-Mlx contributes to the regulation of lipid, carbohydrate and amino acid metabolic genes in mice or flies [20,29,30,31,33,37,41,81]. Organisms can control the intake and excretion of nutrients to maintain nutrient homeostasis. Our results reveal that Mondo-Mlx can control *Tret*, *PFK,* and *PK* involved in blood glucose transport and glycolysis, potentially to balance sugar levels in circulation and to avoid excessive sugar. It is speculated that FAMeT may not be the key enzyme in the JH biosynthesis pathway in silkworms. FAMeT could be related to the absorption and digestion of nutrients in the larval stage, and could be related to the transportation and excretion of metabolic substances in the pupal stage [82]. Mondo-Mlx regulates *FAMet5*, potentially being involved in nutrient absorption and digestion. We also confirmed that lipogenic genes (*FAS* and *ACC*) are regulated by Mondo-Mlx. Glutamine provides biosynthetic materials for cell growth and division. Glutamine is also an essential nitrogen source for the synthesis of non-essential amino acids and nucleotides [74]. Mondo-Mlx is the intermediate linker of sugar and glutamine metabolism by regulating the *GS2* gene. Moreover, we found that not only does *Cabut* establish coordinating nutrition metabolism and circadian clock, but also core the circadian clock gene *Cycle* is regulated by Mondo-Mlx. Mondo-Mlx could play a central role in coordinating the biological processes of nutrient metabolism and circadian rhythm through controlling *Cabut* and *Cycle*. This further confirms the importance of Mondo-Mlx in energy metabolism and the circadian rhythm pathway. To clarify the physiological function of Mondo-Mlx, we established transgenic overexpressed and knocked down strains using a promoter mainly expressed in the fat body. We found that Mondo-Mlx can promote fat accumulation, increase larval, pupal and cocoon shell weight, and the number of eggs in order to improve economic characteristics and reproduction.

In conclusion, through the comprehensive analysis of Mondo-Mlx, we investigated the adaptive molecular mechanism during the domestication of *Mlx*. In domesticated strains, the expression of *Mlx* increased and Mondo-Mlx upregulated genes related to nutrient metabolism pathways. Thus, Mondo-Mlx may promote the metabolic utilization of nutrients for growth and reproduction. Mondo-Mlx can coordinate the nutrition pathway with other pathways such as circadian rhythm by regulating downstream genes in order to maintain the link between a variety of complex physiological processes. There are currently no reports on the functions of Mondo-Mlx in lepidoptera insects. Our research reveals the biological functions of Mondo-Mlx for the first time in silkworms, although bioinformatic analysis has been reported on Mlx and Mondo in *Spodoptera litura*, such as phylogenetic analysis, chromosomal location, and expression profiles [83]. Our work reveals the adaptive mechanism of gene and the exceptional role of gene regulation during the domestication of *Bombyx mori*, as well as providing new targets for improved strains through genetic breeding.

## 4. Materials and Methods

### 4.1. Statistics of Sequence Polymorphism and Neutrality Tests

All of the sequences were combined. The sequences were aligned using DANMAN 6.0 (https://www.lynnon.com/index.html, accessed on 7 February 2024). Polymorphism parameters including S (the number of segregating sites), π (the mean number of nucleotide differences per site), and *θ_w_* (Watterson’s estimator of 4Nem) were analyzed using by DnaSP 5.10 (http://www.ub.edu/dnasp/index_v5.html, accessed on 7 February 2024) [84]. To test deviation from the neutral evolution model and to reveal the evolutionary history between the TE deleted or inserted in domesticated and wild silkworm strains, DnaSP 5.10 was utilized to analyze Tajima’s D test and Fu and Li’s D test.

### 4.2. Cells Culture and Transfection

The BmNs cell line was maintained at 28 °C and cultured in TC-100 insect medium (Gibco, Grand Island, NY, USA) containing 2 billion units per liter of penicillin and streptomycin (Gibco, Grand Island, NY, USA), and 10% (*v*/*v*) fetal bovine serum (Gibco, Grand Island, NY, USA). Cell transfection was conducted with at last 75% cell density. The cells in each well were incubated for 6–8 h in a mixture with 1 μg of plasmid, 2 μL of X-tremeGENE HP DNA Transfection Reagent (Roche, Mannheim, Germany), and TC-100 insect medium serum- and antibiotic-free, then we replaced the medium with serum and antibiotics for 48 h at 28 °C before mRNA extraction, the luciferase assay or the chromatin immunoprecipitation PCR (ChIP-PCR) assay.

### 4.3. Total RNA Extraction and Quantitative Real-Time PCR (qPCR)

Cells were collected after transfecting 48 h, and the total RNA was extracted using TRIzol™ reagent (Invitrogen, Carlsbad, CA, USA). Total RNA was used as templates for reverse transcription reaction. cDNA synthesis was performed to using the reverse transcription kit (Promega, Madison, WI, USA). The primers used for qPCR were listed in the Appendix A. qPCR reactions were conducted in a 20 μL reaction volume with Takara TB Green^®^ Premix Ex Taq™ II (Takara, Dalian, Liaoning, China) by Applied Biosystems7500 (ABI, Carlsbad, CA, USA). The qPCR conditions were 95 °C for 1 min, and 40 cycles of 95 °C for 30 s, 95 °C for 3 s, and 65 °C for 30 s. The last step was a melting curve analysis (from 65 to 95 °C with a slope of 0.1 °C/s). Each reaction was replicated at least three times.

### 4.4. Dual Luciferase Reporter Assay

Promoters of downstream genes were cloned into the pMD-19 vector (Takara, Dalian, Liaoning, China) and then inserted into the pGL3-basic vector (Promega, Madison, WI, USA). The primers used for cloning were listed in the Appendix A. We constructed a shRNA interference vector containing a specific 21 bp fragment initiated by U6 using the method designed by Tanaka [85] (GFP: GAAGGTTATGTACAGGAAAGT, Mlx: GGCTGACGGACCAAGACAAAT, Mondo: GGATCGACTAGCACAAGTAGC). The RNAi vector was a 21 bp shRNA fragment that was ligated into the U6-pEYFP vector (Solarbio, Beijing, China). The sequence and orientation were confirmed via sequencing. The renilla luciferase gene driven by the IE1 promoter was the reference plasmid. The luciferase activity was standardized according to the renilla luciferase activity. At 12 h post-transfection, the cells were treated with DMSO or TSA (Absin, Shanghai, China) at 1 μM for 24 h, then the cells were collected for the luciferase assay. The luciferase activity was conducted to using the Dual-Luciferase^®^ Reporter Assay System (Promega, Madison, WI, USA) after the transfected cells were washed with phosphate-buffered saline (PBS, Beyotime, Shanghai, China). The data were normalized. The ratio of luciferase activity of the control group (shGFP)was defined as 100%, and the experimental group was adjusted by corresponding multiples. All assays were repeated at least three times.

### 4.5. Electrophoretic Mobility Shift Assay (EMSA)

All probes (biotin, cold, and mutation probe) for EMSA were synthesized by Sangon Biotech (Sangon, Shanghai, China). The probes used for EMSA were listed in the Appendix A. The Mlx purified protein fused with the Msyb tag was expressed at 16 °C, with 0.3% (*w*/*v*) IPTG (Beyotime, Shanghai, China). EMSA was performed using the Chemiluminescent EMSA Kit (Beyotime, Shanghai, China). All the binding reactions, including the negative, sample, cold competition, and mutation cold competition reactions, were carried out according to the EMSA kit instructions. The corresponding amount of 5 × EMSA/gel shift binding buffer and nuclease free water, 1 μg of purified protein, 1 μL of the biotin-unlabeled cold or mutation probe (only competition assay), and 1 μL of the biotin-labeled probe were added into a 10 μL reaction volume. To eliminate the non-specific binding of the probe and protein, we mixed the remaining components before adding the biotin-labeled probe and left the mixture to sit at room temperature for 10 min, or allowed the cold probe to react first. Then, we added the biotin-labeled probe, mixed the mixture well and left it to sit at room temperature for 20 min. After adding the loading buffer into the reaction, electrophoresis was performed with 6% polyacrylamide gel on ice at 100 V, then we stopped electrophoresis until the blue dye bromophenol blue in the loading buffer reached 1/4 of the lower edge of the gel. The samples on the gel were transferred to the positively charged nylon membrane. Then, the membrane was cross-linked at 254 nm UV wavelength and 120 MJ/cm^2^ for 45–60 s using the HL-2000-HybriLinker (UVP, Upland, CA, USA). Finally, the biotin-labeled probe was detected via chemiluminescence according to the Super Signal West Pico Chemiluminescent Substrate (Thermo Fisher, Carlsbad, CA, USA).

### 4.6. ChIP-PCR 

The CDS of *Mlx* and *Mondo* were inserted into the A4-p1180 and IE1-pEYFP overexpression plasmid, respectively. BmNs cells were transfected with Mlx and Mondo overexpression plasmids, respectively. After 48 h, cells were collected for ChIP. ChIP was performed according to the manufacturer’s instructions for the ChIP Assay Kit (Beyotime, Shanghai, China). The primers used for ChIP-PCR were listed in the Appendix A. Briefly, cells were incubated with 1% formaldehyde at room temperature for 10 min and the reaction was terminated with glycine solution. PBS containing 1% PMSF (Beyotime, Shanghai, China) was used to clean the cells three times, and then the cells were lysed with ChIP Dilution Buffer. The genome was broken into 250–500 bp fragments via ultrasound. The supernatant was collected via centrifugation and incubated with antibodies at 4 °C overnight. The supernatant was precipitated with protein G magnetic beads (Pierce, Rockford, IL, USA) and purified for PCR detection.

### 4.7. RNA-Seq

When interfering with *Mlx* or *Mondo* in BmNs cells, their expression level was significantly reduced by qRT-PCR. We collected control and RNAi cells with Trizol, and then sent them off for transcriptome sequencing and analysis. RNA-seq was performed with Illumina Hiseq technology with paired-end sequencing. Reads were mapped with Ensemble to the standard *B. mori* reference genome (*Bombyx mori*. ASM15162v1. dna. toplevel. fa). See Appendix A for data analysis.

### 4.8. Transgenic Strains Generated

The 947 bp enhancer Hr3, the 1111 bp promoter of the LP3 gene, CDS with Myc tag of Mlx (or Mondo) and SV40 were amplified. Then, the Hr3, the LP3 promoter, Mlx (or Mondo) and SV40 were orderly conjugated with the pSL1180 vector to form pSL1180-Hr3- [LP3-Mlx (or Mondo)-SV40]. Subsequently, Hr3- [LP3-Mlx (or Mondo)-SV40] was inserted into pBac- [3× P3-Red (or GFP)-SV40] to generate pBac- [3× P3-Red-SV40-Hr3-LP3-Mlx-SV40] and pBac- [3× P3-GFP-SV40-Hr3-LP3-Mondo-SV40]. The primers were listed in the Appendix A. The construction of transgenic interference vectors was similar to the above method, except that the CDS was replaced with a specific inverted repeat fragment. We then microinjected the plasmid into the non-diapause eggs of D9L with germline transformation within 2 h of egg-laying. Next, the microinjected eggs were incubated at room temperature and approximately 75% humidity. The hatched larvae (G0) were fed with fresh mulberry leaves and sibmated after mothing. Lastly, the positive G1 eggs were selected using fluorescence microscope (Zeiss, Oberkochen, Baden-Württemberg, Germany).

### 4.9. Insect Rearing and Statistics of Economic Characteristics

The wild-type and G2 generations were fed with fresh mulberry leaves. Female silkworms were fed separately, and the weight of larvae was investigated every day, with more than 30 silkworms in each group. The pupal weight, whole cocoon weight, cocoon weight, and cocoon shell ratio were measured, and the number of eggs was counted after laying. The female cocoons from the fifth day after pupation were randomly selected. The larval body weight, cocoon shell weight, pupal weight and cocoon shell ratio of WT and transgenic silkworms were measured.

### 4.10. Measure of Triglyceride Content

The fat body on the fifth day of the fifth instar was taken and freeze-dried. The samples of the same mass were weighed and measured for triglyceride content according to the Micro Triglyceride Content Assay Kit (Solarbio, Beijing, China).

### 4.11. Statistical Analysis

Statistical significance was determined with the F test or Student’s *t*-test, * *p* < 0.05; ** *p* < 0.01; *** *p* < 0.001. Statistical data are presented as the means ± SDs of at least three independent biological replicates.

## Figures and Tables

**Figure 1 ijms-25-03427-f001:**
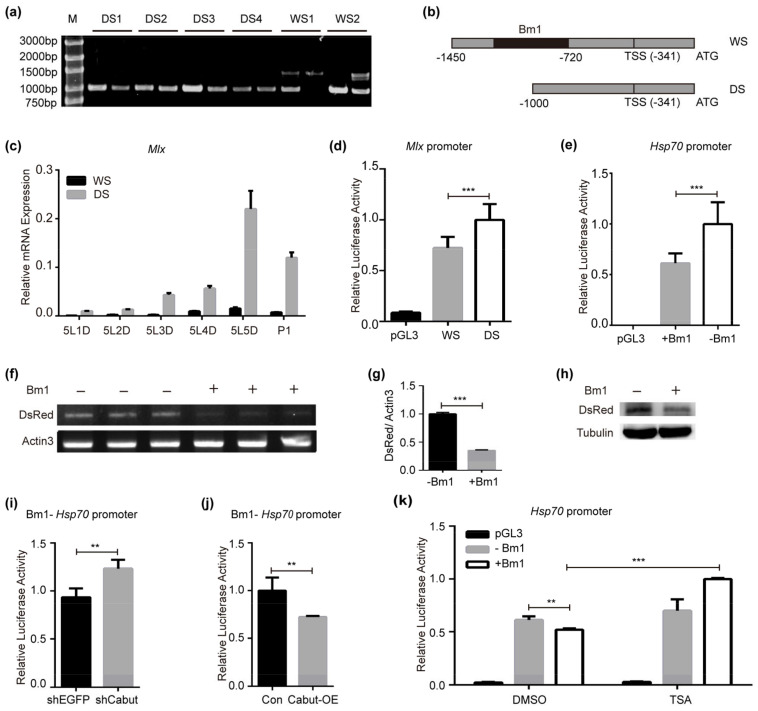
Diversity and activity analyses of *Mlx* upstream regulatory sequence. (**a**) Cloning of *Mlx* upstream regulatory region in domesticated silkworms (DS1-4) and wild silkworms (WS1-2). (**b**) Diagram of *Mlx* upstream regulatory region in domesticated and wild silkworms. (**c**) Comparing the expression of *Mlx* gene in the 5th instar and pupal stage of the fat body in domesticated and wild silkworms. 5L1D: the 1st day of the 5th instar, 5L2D: the 2nd day of the 5th instar, 5L3D: the 3rd day of the 5th instar, 5L4D: the 4th day of the 5th instar, 5L5D: the 5th day of the 5th instar, P1: the 1st day of the pupal stage. (**d**) Activity of *Mlx* upstream regulatory region was examined via a dual-luciferase reporter assay in domesticated and wild silkworms. (**e**) Presence of Bm1 repressed the transcriptional activity of the *Hsp70* promoter. Bm1 was inserted before the luciferase gene driven by the *Hsp 70* promoter. (**f**–**h**) Bm1 decreased the expression of *DsRed* detected using semiquantitative PCR and Western blot. Bm1 was inserted before the *DsRed* gene, driven by the *A4* promoter. (**i**,**j**) Cabut affected the transcriptional repression associated with Bm1. Decreasing the expression of *Cabut* increased the transcriptional activity of the *Hsp 70* promoter when Bm1 was present, increasing the expression of *Cabut* has the opposite effect. (**k**) Treatment with 1 μM TSA can release the transcriptional repression associated with Bm1. Error bars show the SD. Significant differences: ** *p* < 0.01, *** *p* < 0.001.

**Figure 2 ijms-25-03427-f002:**
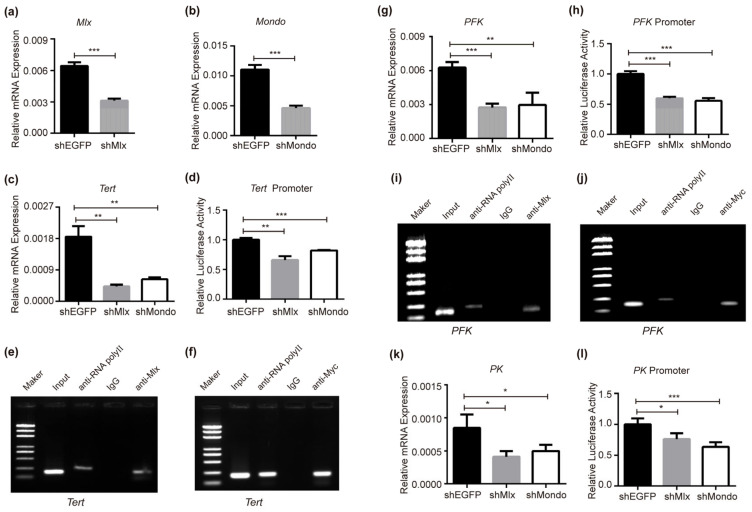
Mlx and Mondo regulate trehalose transport and glycolysis genes. (**a**,**b**) Verifying the interference effects of *Mlx* or *Mondo* in BmNs cells by means of qRT-PCR. (**c**,**d**) Expression and promoter transcriptional activity of *Tret* were reduced via RNAi of *Mlx* or *Mondo* in BmNs cells. (**e**,**f**) Mlx or Mondo being bound to the *Tret* promoter was verified by ChIP-PCR. Specific primers covering the ChoRE element of the *Tret* promoter for ChIP-PCR were designed. Genome fragments were precipitated by Mlx or Myc antibodies after *Mlx* or *Mondo-Myc* overexpression. The IgG antibody was used as negative control and RNA polyII antibody was used as the positive control. (**g**,**h**) Expression and promoter transcriptional activity of *PFK* were decreased via RNAi of *Mlx* or *Mondo* in BmNs cells. (**i**,**j**) ChIP-PCR confirmed Mlx or Mondo being bound to *PFK* promoter. See above for experimental details. (**k**,**l**) RNAi of *Mlx* or *Mondo* decreased the expression and the promoter activity of *PK*. Error bars show SD. Significant differences: * *p* < 0.05, ** *p* < 0.01, *** *p* < 0.001.

**Figure 3 ijms-25-03427-f003:**
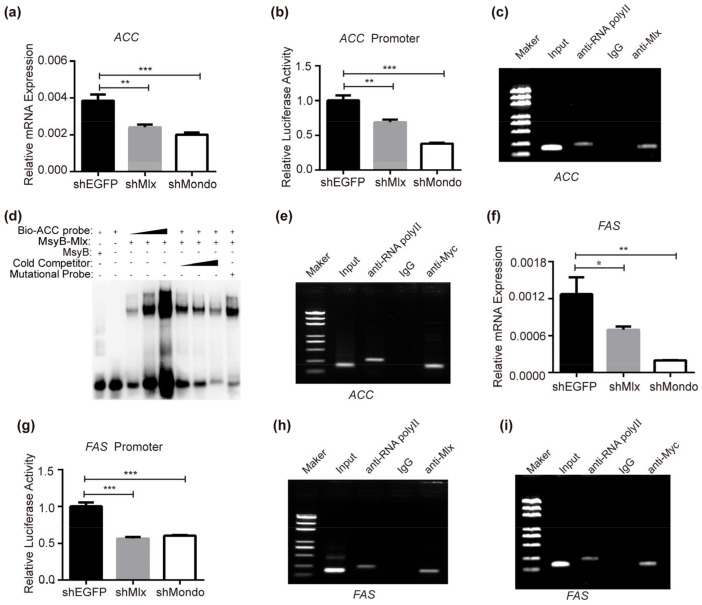
Mondo-Mlx regulates the expression of lipogenic genes. (**a**,**b**) RNAi of *Mlx* or *Mondo* reduced the expression and promoter transcriptional activity of *ACC* in BmNs cells. (**c**,**d**) ChIP-PCR and ENSA with *Mlx* overexpression in BmNs cells. Specific primers covering the ChoRE element of the *ACC* promoter were used for ChIP-PCR. Prokaryotically expressed Mlx-MsyB fusion protein was purified, and specific biotinylated probe based on the ChoRE element of the *ACC* promoter for EMSA was designed. (**e**) ChIP-PCR assay in BmNs cells with *Mondo*-*Myc* overexpression. Specific primers covering ChoRE element of the *FAS* promoter were used. (**f**,**g**) Expression and promoter transcriptional activity of *FAS* was decreased when *Mlx* or *Mondo* interfered with the shRNA in BmNs cells. (**h**,**i**) ChIP-PCR assay of the direct binding of *Mlx* or *Mondo* to the ChoRE element of the *FAS* promoter in BmNs cells. Error bars show SD. Significant difference: * *p* < 0.05, ** *p* < 0.01, *** *p* < 0.001.

**Figure 4 ijms-25-03427-f004:**
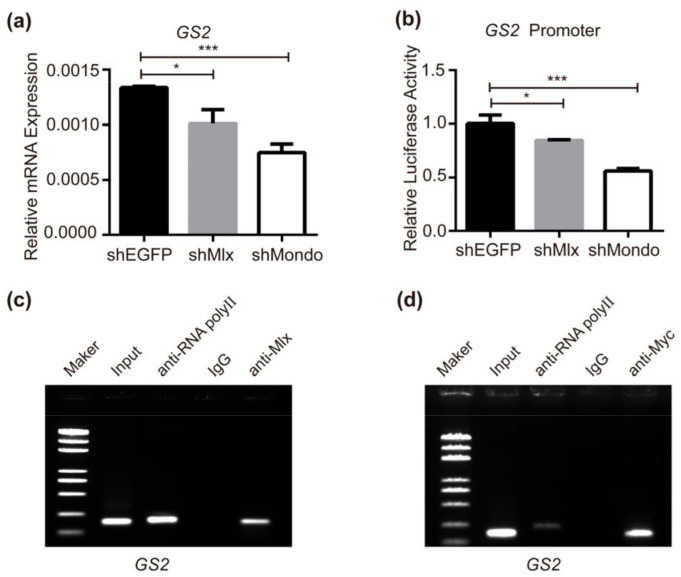
Mlx and Mondo are involved in regulation of *GS2* expression. (**a**,**b**) RNAi of *Mlx* or *Mondo* reduced the expression and promoter transcriptional activity of *GS2* in BmNs cells. (**c**,**d**) Binding of Mlx or Mondo to the ChoRE element of the *GS2* promoter was examined by means of ChIP-PCR. Error bars show SD. Significant differences: * *p* < 0.05, *** *p* < 0.001.

**Figure 5 ijms-25-03427-f005:**
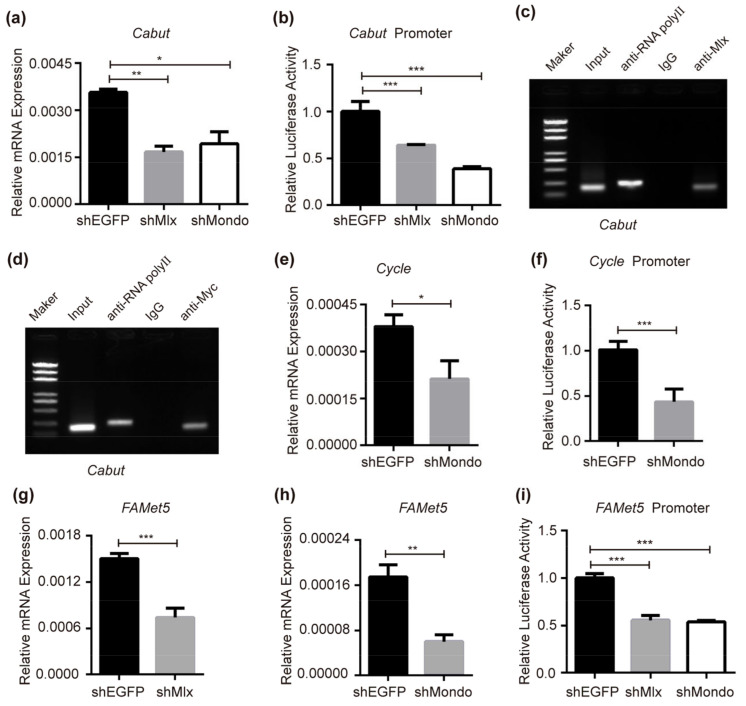
Effects of Mlx and Mondo on *Cabut*, *Cycle* and *FAMeT5*. (**a**) Knock down of *Mlx* or *Mondo* repressed the expression of *Cabut* in BmNs cells. (**b**) When knocking down *Mlx* or *Mondo* in BmNs cells, the transcriptional activity of *Cabut* promoter was inhibited. (**c**,**d**) Binding of Mlx or Mondo to the *Cabut* promoter was examined by means of ChIP-PCR. (**e**) Decreased expression of *Mondo* led to reduced expression of *Cycle* in BmNs cells. (**f**) Transcriptional activity of *Cycle* promoter was inhibited when *Mondo* was reduced in BmNs cells. (**g**,**h**) Expression of *FAMeT5* decreased after the knock down of *Mlx* or *Mondo* in BmNs cells. (**i**) Knock down of *Mlx* or *Mondo* repressed *FAMeT5* promoter activity in BmNs cells. Error bars show SDs. Significant differences: * *p* < 0.05, ** *p* < 0.01, *** *p* < 0.001.

**Figure 6 ijms-25-03427-f006:**
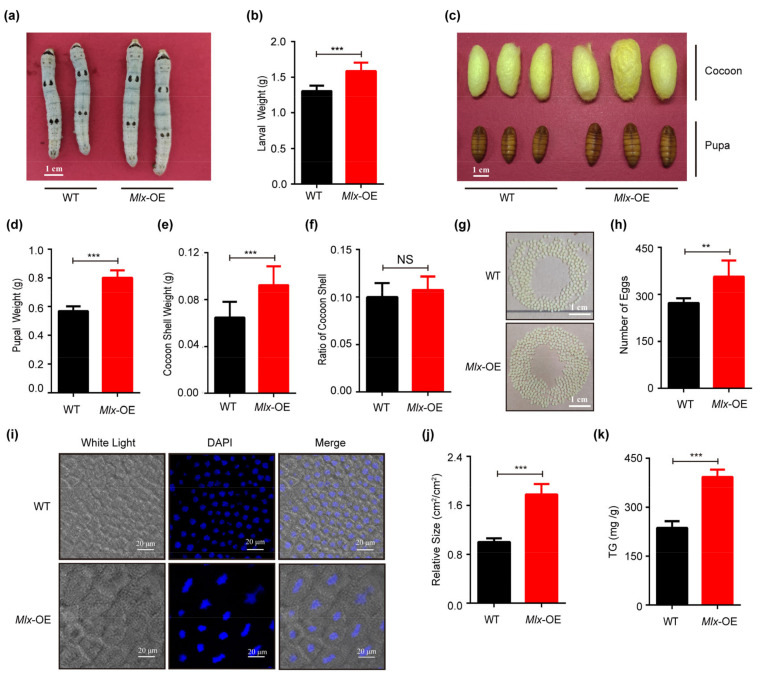
Effects of overexpression of *Mlx* (*Mlx*-OE) in transgenic silkworms. (**a**,**b**) Comparison of body size and larval weight of wild-type silkworm (WT) and *Mlx*-OE on the fifth day of fifth instar. (**c**–**h**): Investigation of pupal and cocoon size (**c**), pupal weight (**d**), cocoon shell weight (**e**), cocoon shell ratio (**f**), and egg number between (**g**,**h**) *Mlx*-OE and WT. (**i**–**k**) Analysis of size (**i**,**j**) and triglyceride content (**k**) of fat body cells in *Mlx*-OE and WT on the fifth day of the fifth instar. Error bars show SD. Significant differences: ** *p* < 0.01; *** *p* < 0.001; NS, no significant difference.

**Figure 7 ijms-25-03427-f007:**
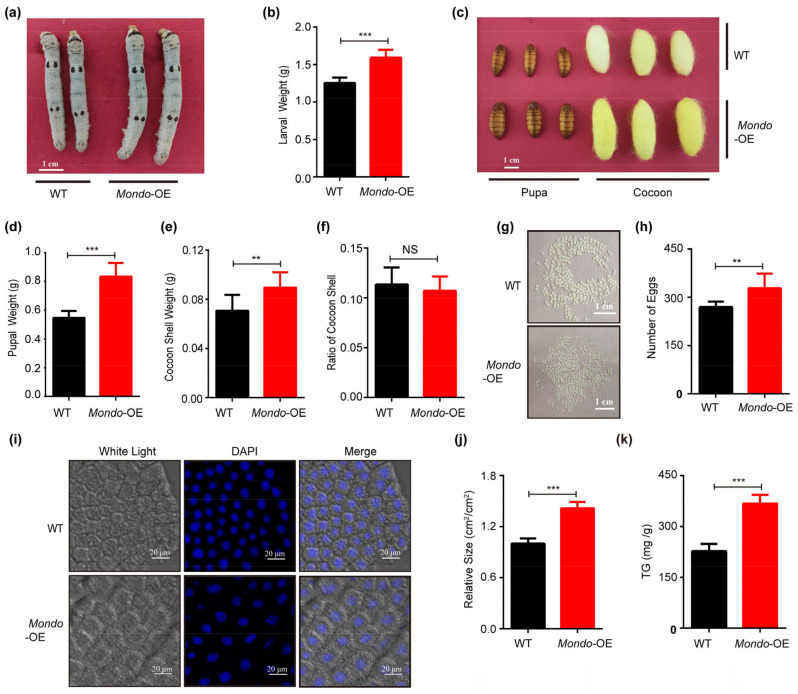
Phenotypic analysis of *Mondo* overexpression (*Mondo*-OE) in transgenic silkworms. (**a**–**h**) Investigation of larval growth and body weight (**a**,**b**), pupal and cocoon size (**c**), pupal weight, cocoon shell weight and ratio (**e**,**f**), and egg number (**g**,**h**) of *Mondo*-OE and WT. (**i**–**k**) Detection of size (**i**,**j**) and triglyceride content (**k**) of fat body cells in *Mondo*-OE and WT on the fifth day of the fifth instar. Error bars show SDs. Significant differences: ** *p* < 0.01; *** *p* < 0.001; NS, no significant difference.

**Figure 8 ijms-25-03427-f008:**
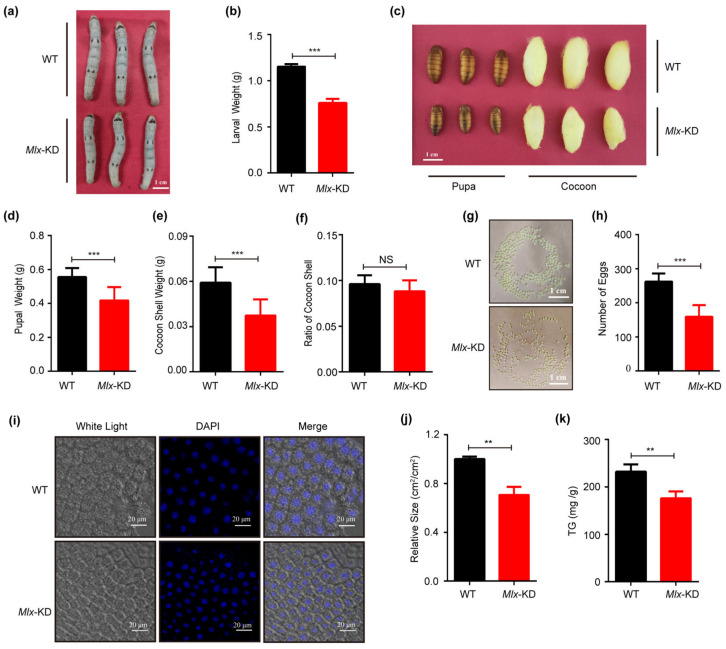
Effects of *Mlx* knock-down (*Mlx*-KD) in transgenic silkworms. (**a**–**h**): Analysis of larval size and weight (**a**,**b**), pupal and cocoon size (**c**), pupal weight (**d**), cocoon shell weight and ratio (**e**,**f**), and egg number (**g**,**h**) between *Mlx*-KD and WT. (**i**–**k**) Test of size (**i**,**j**) and triglyceride content (**k**) of fat body cells in *Mlx*-KD and WT on the fifth day of the fifth instar. Error bars show SD. Significant differences: ** *p* < 0.01; *** *p* < 0.001; NS, no significant difference.

**Figure 9 ijms-25-03427-f009:**
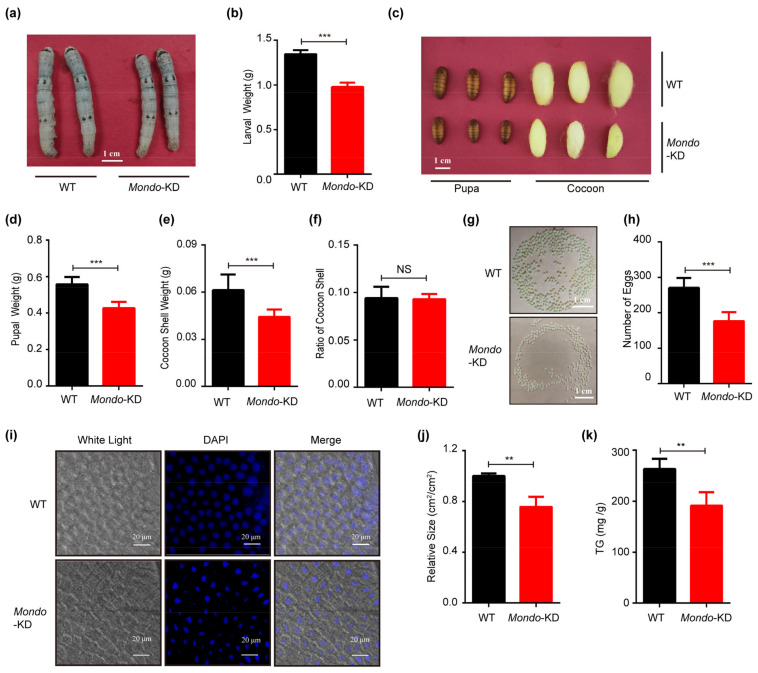
Interference effect of *Mondo* (*Mondo*-KD) in transgenic silkworms. (**a**–**h**) Larval size and weight (**a**,**b**), pupal and cocoon size (**c**), pupal weight (**d**), cocoon shell weight and ratio (**e**,**f**), and egg number (**g**,**h**) between *Mondo*-KD and WT. (**i**–**k**) Measurement of size (**i**,**j**) and triglyceride content (**k**) of fat body cells in *Mondo*-KD and WT on the fifth day of the fifth instar. Error bars show SDs. Significant differences: ** *p* < 0.01; *** *p* < 0.001; NS, no significant difference.

**Table 1 ijms-25-03427-t001:** Nucleotide diversity and neutral test.

Test Set	5′-Flanking Region (~3 kb)					
	S	π	*θ* *w*	Tajima’s D	Fu and Li’s D	Fu and Li’s F
Domestic (*n* = 18)	232	0.00842	0.02202	−2.62506	−4.00135	−4.1804
Wild (*n* = 8)	123	0.01751	0.01622	0.34000	0.97307	0.92106
With TE (*n* = 4)	377	0.06996	0.06066	1.5464	1.56333	1.68282
Without TE (*n* = 4)	126	0.02301	0.02326	−0.35185	−0.27742	−0.31234

## Data Availability

Data are contained within the article and Appendix A.

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
