# Peer review of "Domestication Gene *Mlx* and Its Partner *Mondo* Are Involved in Controlling the Larval Body Size and Cocoon Shell Weight of *Bombyx mori"

_ijms, 2024, doi:10.3390/ijms25063427_

Round 1
Reviewer 1 Report
Comments and Suggestions for Authors
The manuscript by Qin et al is an excellent example of thorough investigation of phenotypic basis by forward genetics. Here, they are investigating the potential for a transposable element (Bm1, a putative SINE) in the upstream CRE of the Mlx coding sequence to affect transcript and physiological outputs of metabolic, nutrient and ultimately developmental pathways in wild type vs domestic silk moths. Their experiment design is top notch with proper controls apparent in all assays. Consequently, the data are unambiguous and appear to support the hypothesis that the loss of Bm1, and thus release of expression of energy accumulating pathways and thus increased growth potential, is associated with the increased silk performance of domesticated bombyx as compared to wild type. Given the strength fo the experiment design and activity, there are no major concerns and only minor concerns:
· Lines 33-34: what does “completely domesticated artificially lepidopteran” mean?
· 155-6: the authors claim that the results “showed that Bm1 inhibited the transcriptional activity…”, however their methods are not designed to show Bm1 inhibits; rather, the absence of Bm1 is associated with higher transcript numbers while its presence is associated with lower. More work must be done to demonstrate Bm1 inhibits.
· 169: authors should provide rationale and basis for Trichostatin use and interpretation of basis for results.
· Fig 1A: why is there no lower band in W1 lane 2, and no upper in lane 1 W2?
· Section 2.3 and Fig 2: is it Tret or Tert?
· 212: Drosophila is a genus and thus should be italicized and capitalized
· Section 4.4: given the immense role that the dual luc reporter assay plays in this study, more details are needed both int eh methods but especially in figures to communicate what constitutes controls and ratio (given relative LUC levels are given in several)
Comments on the Quality of English LanguageExtensive english editing is required: while the content and message is generally decipherable, editing will improve readability and ensure accuracy
Author Response
Response to Reviewer 1 Comments
Summary
We sincerely thank the editor and the reviewers for making constructive remarks and useful suggestions, which have significantly raised the quality of the manuscript and have enabled us to improve the manuscript. We have made revisions based on reviewers’ comments. Please find the detailed responses below and the corresponding revisions highlighted in the re-submitted files. To improve readability and ensure accuracy, our manuscript have undergone extensive English revisions. We would like to use the English editing services of MDPI after our manuscript has been accepted by IJMS, if extensive English revisions are not approved by the editor. If there are any incorrect answers or questions in the manuscript, please do not hesitate to let us know.
Point by point response to Comments and Suggestions for Authors
Comments 1: Lines 33-34: what does “completely domesticated artificially lepidopteran” mean?
Response 1: Thank you for pointing this out. This sentence has revised as “The domesticated silkworm, Bombyx mori, is the only truly domesticated lepidopteran insect, which has important economic value.” Please see lines 34.
Comments 2: 155-6: the authors claim that the results “showed that Bm1 inhibited the transcriptional activity…”, however their methods are not designed to show Bm1 inhibits; rather, the absence of Bm1 is associated with higher transcript numbers while its presence is associated with lower. More work must be done to demonstrate Bm1 inhibits.
Response 2: Thanks for this constructive comment. Due to our current weak results, we will refer to your comment to revise these results in our manuscript. These results should be revised as “The absence of Bm1 is associated with higher transcriptional activity of Hsp70 promoter, while its presence is associated with lower (Figure 1e). Moreover, this association is also supported by elevated transcription and translation of DsRed (Figure 1f-h). Results showed that presence of Bm1 is associated with transcriptional repression.” Please see lines 159-163.
Comments 3: 169: authors should provide rationale and basis for Trichostatin use and interpretation of basis for results.
Response 3: Thank you for this constructive comment. We will refer to your comment to provide rationale and basis for TSA and interpret these results in our manuscript. Firstly, acetylation and deacetylation modifications of histones can affect gene expression. HAT acetylates the histones, forming an "open" chromatin structure to facilitate transcription; in contrast, HDAC deacetylates histones and forms a "closed" structure in chromatin, leading to gene silencing. (References: Grunstein, 1997; Eberharter and Becker, 2002; Vidal and Gaber, 1991; Yang et al.,1996, 1997; Kao et al., 2000,2002; Zhou et al., 2000). Secondly, Trichostatin A (TSA) is a commonly used HDAC (histone deacetylase) inhibitor. TSA can strongly, and selectively inhibit HDAC. Inhibition of HDAC by TSA can increase histone acetylation levels and lead to increase transcription levels of some genes. (References: Yoshida, et al.,1990; Gräff, et al., 2013). The higher derepression upon TSA treatment under Bm1 presence in the upstream regulatory region supports our hypothesis that additional deacetylation in Bm1 affects gene expression level. Please see lines 175-184.
Comments 4: Fig 1A: why is there no lower band in W1 lane 2, and no upper in lane 1 W2?
Response 4: Thank you for pointing this out. The PCR template (genome) for each lane is derived from individual sample. We speculate that if the genome is homozygous in wild silkworm, there is only upper band; when the genome is heterozygous, lower and upper bands will appear. We added this interpretation in the main text. Please See lines 124-126.
Comments 5: Section 2.3 and Fig 2: is it Tret or Tert?
Response 5: Thank you for pointing this out. It should be “Tret”. We have revised in our manuscript.
Comments 6: 212: Drosophila is a genus and thus should be italicized and capitalized
Response 6: Thank you for pointing this out. We have revised in the manuscript.
Comments 7: Section 4.4: given the immense role that the dual luc reporter assay plays in this study, more details are needed both int eh methods but especially in figures to communicate what constitutes controls and ratio (given relative LUC levels are given in several)
Response 7: Thank you for this constructive comment. We have revised in materials and methods 4.4. Please see lines 469-470 and 475-477.
Comments on the Quality of English Language
Comments 8: Extensive english editing is required: while the content and message is generally decipherable, editing will improve readability and ensure accuracy
Response 8: We thank you for the positive comments on our manuscript. The manuscript was further checked by a colleague fluent in English writing.

Reviewer 2 Report
Comments and Suggestions for Authors
On request of IJMS, I have revised the manuscript titled “Domestication gene Mlx and its partner Mondo are involved in controlling larval body and cocoon shell weight of Bombyx mori”, by Xiaoxuan Qin et al.
The main scope of this work was clarifying the molecular mechanism underlying the improvement of economically relevant characteristics of Bombyx mori silkworm, including larval body and cocoon shell weight, during domestication. To this end, by comprehensive analysis of Mondo-Mlx complex, the authors evidenced that upon domestication, while the expression of Mlx increased, Mondo-Mlx upregulated genes related to nutrient metabolism pathways, thus promoting utilization of nutrients for growth and reproduction, as well as coordinated the nutrition pathway with other paths such as circadian rhythm.
General Comments
This work revealed the adaptive mechanism of gene and the exceptional role of gene regulation during domestication of domesticated silkworm, thus providing new targets for improved strains by genetic breeding. Due to the economical relevance of silkworms and silk production worldwide, the findings reported in this study could be of great interest.
The paper is generally well written, and English is fine.
Anyway, there are few minor issues, regarding especially the format, that should be addressed to allow the publication of this study on IJMS.
Minor issues
Line 13. Please, remove “A”, and rephrase the sentence in lines 13-14.
In the manuscript there are several issues regarding the way by which words have been written so that the paper format results not uniform. Italic or/and capital letter are used without precise rules. As examples, along all manuscript authors should make uniform the way of writing MLX and Mondo. Please, compare lines 2 and 19 with lines 23 and 24, or also with lines 55 and 56 and so on. The same for name of enzymes. Italic or not italic? Please, check carefully all the paper to solve similar issues.
Please, carefully check all paper and specify the abbreviations at their first mention in all sections, where necessary.
Please, for all reagents and instruments, authors should provide the manufacturer name (producer), its city and country. Please, check the manuscript and provide all information where missing.
Line 543. Since authors have submitted Supplementary Materials, they should insert here a list of its contents. Example: Figure S1: Caption. Figure S2: Caption…
The reference list does not respect the requirements of IJMS. Please, reformat it according to the instructions of IJMS.
I suggest IJMS, to consider this article for publication after authors will have addressed the above mentioned minor issues.
Author Response
Response to Reviewer 2 Comments
Summary
We sincerely thank the editor and the reviewers for making constructive remarks and useful suggestions, which have significantly raised the quality of the manuscript and have enabled us to improve the manuscript. We have made revisions based on reviewers’ comments. Please find the detailed responses below and the corresponding revisions highlighted in the re-submitted files. If there are any incorrect answers or questions in the manuscript, please do not hesitate to let us know.
Point by point response to Comments and Suggestions for Authors
Comments 1: Line 13. Please, remove “A”, and rephrase the sentence in lines 13-14.
Response 1: Thank you for pointing this out. We have removed “A”. And we have revised as “The Bombyx mori was domesticated from Bombyx mandarina”. Please see lines 16.
Comments 2: In the manuscript there are several issues regarding the way by which words have been written so that the paper format results not uniform. Italic or/and capital letter are used without precise rules. As examples, along all manuscript authors should make uniform the way of writing MLX and Mondo. Please, compare lines 2 and 19 with lines 23 and 24, or also with lines 55 and 56 and so on. The same for name of enzymes. Italic or not italic? Please, check carefully all the paper to solve similar issues.
Response 2: Thank you very much for your comments. We have checked and revised in our manuscript. Based on the reported literatures, we found that Mlx also was written as MLX or Bigmax etc, and Mondo also was written as ChREBP, MLXIPL or Mlxipl etc. We have uniformly revised Mlx and Mondo in the main text. We used italic when it represents genes, and used regular when it represents protein.
Comments 3: Please, carefully check all paper and specify the abbreviations at their first mention in all sections, where necessary.
Response 3: Thank you very much for your advice. we have checked all abbreviations at their first mention throughout the manuscript.
Comments 4: Please, for all reagents and instruments, authors should provide the manufacturer name (producer), its city and country. Please, check the manuscript and provide all information where missing.
Response 4: Thank you very much for your advice. we have checked and provide all information for all reagents and instruments.
Comments 5: Line 543. Since authors have submitted Supplementary Materials, they should insert here a list of its contents. Example: Figure S1: Caption. Figure S2: Caption…
Response 5: Thank you very much for your advice. We have revised in our manuscript. Please see lines 550-571.
Comments 6: The reference list does not respect the requirements of IJMS. Please, reformat it according to the instructions of IJMS.
Response 6: Thank you very much for your advice. we have reformatted references. Please see lines 584-777.
Reviewer 3 Report
Comments and Suggestions for Authors
The article deals with the economically extremely important species that is the silkworm. Admittedly, the Mlx gene has already been analyzed in this species (there are several papers available in the literature), but the authors study it in a new context - that is, the domestication of the silkworm.
And it is the comparison of the wild form with the domesticated silkworm that proves the value of the work.
In terms of molecular methods and their use, the article is correct. On the other hand, I would recommend that the authors use another statistical test in addition to the Student's t-test!
I have reservations about using the names of species and genera in the work. Authors should write them using italics. Please correct both in the main text and references.
Has the Mlx gene been analyzed in other representatives of butterflies, if there is literature data (https://onlinelibrary.wiley.com/doi/full/10.1111/1744-7917.12979) then please compare these results.
Author Response
Response to Reviewer 3 Comments
Summary
We sincerely thank the editor and the reviewers for making constructive remarks and useful suggestions, which have significantly raised the quality of the manuscript and have enabled us to improve the manuscript. We have made revisions based on reviewers’ comments. Please find the detailed responses below and the corresponding revisions highlighted in the re-submitted files. If there are any incorrect answers or questions in the manuscript, please do not hesitate to let us know.
Point by point response to Comments and Suggestions for Authors
Comments 1: In terms of molecular methods and their use, the article is correct. On the other hand, I would recommend that the authors use another statistical test in addition to the Student's t-test!
Response 1: Thank you very much for your comment. We also used F test.
Comments 2: I have reservations about using the names of species and genera in the work. Authors should write them using italics. Please correct both in the main text and references.
Response 2: Thank you very much for your comment. We have corrected the format of species and genera in the main text and references.
Comments 3: Has the Mlx gene been analyzed in other representatives of butterflies, if there is literature data (https://onlinelibrary.wiley.com/doi/full/10.1111/1744-7917.12979) then please compare these results.
Response 3: Thank you very much for your comment. We have compared these results and added this reference in our manuscript. Please see lines 426-430.